# ARL13B regulates Sonic hedgehog signaling from outside primary cilia

Eduardo D Gigante[1,2], Megan R Taylor[3], Anna A Ivanova[4], Richard A Kahn[4], Tamara Caspary[2]*

[1]Neuroscience Graduate Program, Emory University School of Medicine, Atlanta, United States; [2]Department of Human Genetics, Emory University School of Medicine, Atlanta, United States; [3]Emory College of Arts and Sciences, Emory University School of Medicine, Atlanta, United States; [4]Department of Biochemistry, Emory University School of Medicine, Atlanta, United States

**Abstract** ARL13B is a regulatory GTPase highly enriched in cilia. Complete loss of *Arl13b* disrupts cilia architecture, protein trafficking and Sonic hedgehog signaling. To determine whether ARL13B is required within cilia, we knocked in a cilia-excluded variant of ARL13B (V358A) and showed it retains all known biochemical function. We found that ARL13B$^{V358A}$ protein was expressed but could not be detected in cilia, even when retrograde ciliary transport was blocked. We showed *Arl13b$^{V358A/V358A}$* mice are viable and fertile with normal Shh signal transduction. However, in contrast to wild type cilia, *Arl13b$^{V358A/V358A}$* cells displayed short cilia and lacked ciliary ARL3 and INPP5E. These data indicate that ARL13B's role within cilia can be uncoupled from its function outside of cilia. Furthermore, these data imply that the cilia defects upon complete absence of ARL13B do not underlie the alterations in Shh transduction, which is unexpected given the requirement of cilia for Shh transduction.

## Introduction

The Hedgehog (Hh) signaling pathway is essential for embryogenesis in a wide variety of organisms. Initially discovered in *Drosophila* where there is a single Hh ligand, the core components of the Hh pathway are conserved in vertebrates (*Nüsslein-Volhard and Wieschaus, 1980*). These include the vertebrate Hh receptor Patched1 (Ptch1), the obligate transducer of the pathway Smoothened (Smo), as well as the Gli transcription factors (Ci in *Drosophila*) that act as both activators (GliA) and repressors (GliR) to control target gene transcription (*Briscoe and Thérond, 2013*). There are three Hh ligands in vertebrates, Sonic (Shh), Indian (Ihh) and Desert (Dhh), that regulate a multitude of developmental processes including formation of the limbs and digits, the bones of the skull and face, and the patterning of the neural tube (*Placzek and Briscoe, 2018*). Diminished Hh signaling during embryogenesis results in birth defects whereas increased Hh signaling leads to tumors, highlighting the importance of the pathway and its regulation (*Raleigh and Reiter, 2019*).

Given the deep homology between invertebrate and vertebrate Hh signaling, the discovery that primary cilia are required for vertebrate, but not invertebrate, Hh signaling was unexpected (*Huangfu et al., 2003*; *Huangfu and Anderson, 2006*). Vertebrate Hh components dynamically traffic within primary cilia in response to Hh ligand (*Corbit et al., 2005*; *Haycraft et al., 2005*; *Rohatgi et al., 2007*). In the absence of ligand, Ptch1 is enriched on the ciliary membrane and Smo is barely detectable (*Corbit et al., 2005*; *Rohatgi et al., 2007*). Furthermore, full length Gli proteins traffic to the ciliary tip and back to the cytoplasm before being cleaved to their repressor form, which actively shuts down Hh target gene transcription in the nucleus (*Kim et al., 2009*; *Liu et al., 2005*; *Wen et al., 2010*; *Humke et al., 2010*). In contrast, upon ligand stimulation (Shh, Ihh or Dhh), the Ptch1 receptor binds the ligand and shuttles out of cilia (*Rohatgi et al., 2007*). Subsequently,

*For correspondence:
tcaspar@emory.edu

Competing interests: The authors declare that no competing interests exist.

Smo is enriched in the cilium and is subsequently activated (*Kong et al., 2019*; *Corbit et al., 2005*). Activated Smo promotes the processing of full-length Gli transcription factors into GliA, which turns on target genes (*Aza-Blanc et al., 2000*; *Ruiz i Altaba, 1998*). Ablation of cilia results in an absence of GliA and GliR production, rendering the pathway inert and leading to an absence of transcriptional response (*Huangfu and Anderson, 2005*). The dynamic ciliary movement of Shh components appears to be critical to pathway function, as alterations to cilia disrupt pathway output (*Caspary et al., 2007*; *Huangfu and Anderson, 2006*; *Liem et al., 2012*; *Liem et al., 2009*; *Goetz et al., 2009*; *Murdoch and Copp, 2010*; *Tuz et al., 2014*; *Cortellino et al., 2009*; *Houde et al., 2006*; *Liu et al., 2005*; *Tran et al., 2008*; *Taylor et al., 2015*).

Given that the fundamental logic of the pathway is conserved from flies through vertebrates, and that flies transduce Hh signals without relying on primary cilia, both evolutionary and mechanistic questions are raised as to how vertebrate cells co-opted the primary cilium for Hh signal transduction. One distinction lies in the fact that vertebrates use Hh signaling over a longer distance than flies, leading to the proposal that the primary cilium is a critical part of a mechanism underlying long range signaling (*Bangs et al., 2015*). For example, in neural patterning Shh is initially expressed in the notochord and is secreted to specify fates more than 20 cells away (*Chiang et al., 1996*; *Briscoe et al., 2001*; *Roelink et al., 1994*). At the evolutionary level, comparisons among organisms with cilia and or Hh have provided some clues. The round worm *C. elegans* have cilia yet do not possess Hh signaling as they don't have most of the genes encoding the core components of Hh signal transduction (*The C. elegans Sequencing Consortium, 1998*; *Roy, 2012*). Curiously, a few components of Hh signaling such as fused and costal2 are in the *C. elegans* genome where they are functionally important for ciliogenesis (*Ingham et al., 2011*). Additionally, *C. elegans* retained a Ptch1 homolog important for development and pattern formation, but no Hh or Smo (*Zugasti et al., 2005*; *Kuwabara et al., 2000*). In contrast, planaria flatworms possess both cilia and Hh signaling but the cilia are not required to transduce Hh signaling (*Rink et al., 2009*). The first known evolutionary link between cilia and Hh is in sea urchins which transduce Hh signal in developing muscle tissue via motile cilia (*Warner et al., 2014*; *Sigg et al., 2017*). Subsequently, in vertebrates Hh signaling requires primary cilia. These data suggest that the mechanistic link of cilia and Hh is limited to deuterostomes and raises the question of whether the relationship of Hh and primary cilia originated near the last common ancestor of vertebrates, the urochordates.

ARL13B is a member of the ARF family of regulatory GTPases and is highly enriched on the ciliary membrane (*Caspary et al., 2007*). In mice, a null mutation of *Arl13b* leads to short cilia and to alterations in Shh signal transduction (*Caspary et al., 2007*; *Larkins et al., 2011*). ARL13 is ancient, predicted to be present in the last common eukaryotic ancestor. ARL13 appears to have been lost during evolution in organisms that lack cilia and duplicated to ARL13A and ARL13B in the urochordates, thus ARL13B is proposed to hold important clues in deciphering the links between primary cilia and vertebrate Hh signaling (*Schlacht et al., 2013*; *Li et al., 2004*; *Kahn et al., 2008*; *East et al., 2012*; *Logsdon, 2004*).

ARF regulatory GTPases, like ARL13B, are best known to play roles in membrane trafficking (*D'Souza-Schorey and Chavrier, 2006*). As is true for a large number of regulatory GTPases, ARL13B is functionally diverse (*Sztul et al., 2019*). It regulates endocytic traffic (*Barral et al., 2012*), as well as the phospholipid composition of the ciliary membrane through recruitment of the lipid phosphatase INPP5E to the ciliary membrane (*Humbert et al., 2012*). ARL13B also has a conserved role as a guanine nucleotide exchange factor (GEF) for ARL3, another ciliary ARF-like (ARL) protein (*Gotthardt et al., 2015*; *Zhang et al., 2016*; *Hanke-Gogokhia et al., 2016*; *Ivanova et al., 2017*). ARL13B regulates intraflagellar transport (IFT), the process that builds and maintains cilia (*Cevik et al., 2010*; *Li et al., 2010*; *Nozaki et al., 2017*). It is known to interact with several proteins associated with cilia, including the exocyst, tubulin and UNC119 (*Seixas et al., 2016*; *Zhang et al., 2016*; *Larkins et al., 2011*; *Revenkova et al., 2018*). Critical to this work, loss of ARL13B disrupts Shh signal transduction in at least two distinct ways: Smo enrichment in cilia occurs even in the absence of ligand and Gli activator production is diminished, although Gli repressor is made normally (*Caspary et al., 2007*; *Larkins et al., 2011*).

Due to the high enrichment of ARL13B on the ciliary membrane, ARL13B is assumed to function in its diverse roles from within the cilium. However, ARL13B is present in early endosomes and circular dorsal ruffles on the cell surface (*Barral et al., 2012*; *Casalou et al., 2014*). We previously showed that a V358A variant of ARL13B does not localize to cilia as it disrupts a known VxPx cilia localization

sequence (*Higginbotham et al., 2012*; *Mariani et al., 2016*). Exogenous overexpression of a ARL13B[V358A] construct in *Arl13b* null cells does not rescue ARL13B-dependent phenotypes such as cilia length as well as interneuron migration and connectivity, consistent with ciliary ARL13B mediating these processes (*Higginbotham et al., 2012*; *Mariani et al., 2016*). In contrast, we found that overexpressed ARL13B[V358A] does rescue the Shh-dependent ciliary enrichment of Smo in mouse embryonic fibroblasts, arguing that ARL13B may function outside the cilium to regulate Smo traffic (*Mariani et al., 2016*). Together, these results raise the question of where ARL13B functions.

To define where ARL13B functions in relation to cilia, we wanted an *in vivo* model so generated mice carrying the ARL13B[V358A] point mutation using CRISPR/Cas9. Here we demonstrate that ARL13B[V358A] protein was undetectable in *Arl13b[V358A/V358A]* cilia in both neural tube and mouse embryonic fibroblast cilia, even after blocking retrograde ciliary traffic. We report that *Arl13b[V358A/V358A]* mice were viable, fertile, and transduced Shh signal normally. We found ARL3 and INPP5E did not localize to the short cilia present in/on *Arl13b[V358A/V358A]* cells. These data indicate that ARL13B's roles within and outside cilia can be uncoupled; ARL13B's role in regulating cilia length is from within cilia, whereas its control of Shh signaling appears to be from outside the cilium. Thus, these data imply that the cilia defects seen in the complete absence of ARL13B do not underlie the alterations in Shh transduction, which is unexpected given the requirement of cilia for Shh signal transduction.

## Results

### ARL13B[V358A] displays normal GEF activity

We previously showed that mouse ARL13B[V358A] protein retained normal intrinsic and GAP-stimulated GTP hydrolysis activities by analyzing GST-ARL13B[V358A] purified from human embryonic kidney (HEK) cells (*Mariani et al., 2016*). In order to test ARL13B[V358A] GEF activity for ARL3, we used the same GST-ARL13B[V358A] protein preparation and measured the rates of spontaneous or GEF-stimulated GDP dissociation from ARL3 in the presence and absence of ARL13B or ARL13B[V358A]. ARL3 spontaneously releases pre-bound GDP quite slowly under the conditions in the assay, though release is linear with time, and by extrapolation requires just under 30 minutes for 50% of pre-bound [3H]GDP to dissociate (*Figure 1*). In marked contrast, addition of ARL13B under the same conditions caused the release of 50% of the GDP within ~10 seconds and close to 100% in one minute. We detected no differences between the wild type and mutant ARL13B[V358A] proteins in this assay,

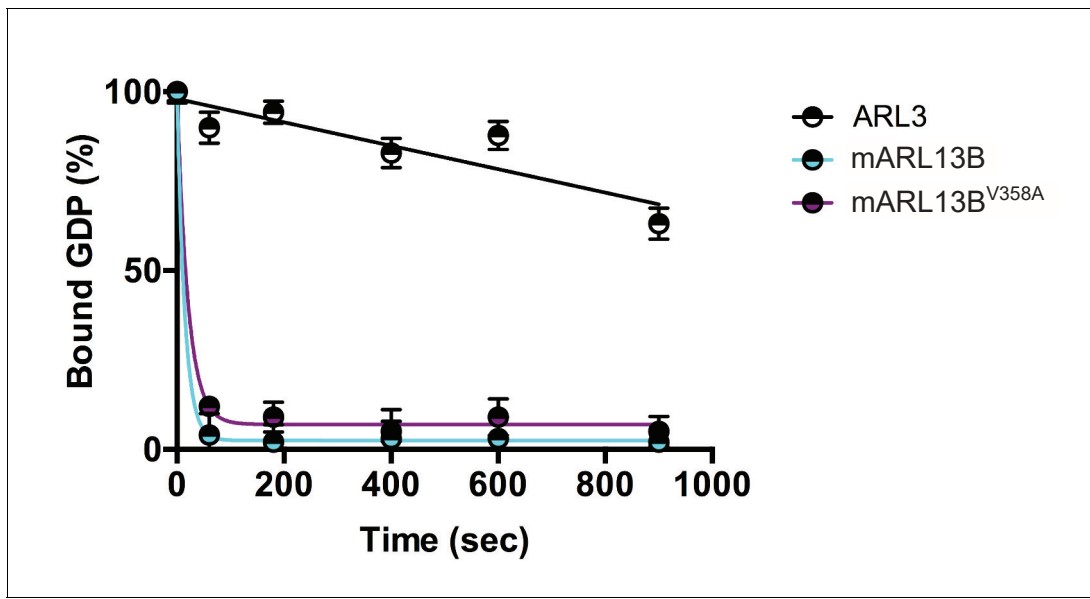

**Figure 1.** ARL3 GEF activity is retained in the ARL13B[V358A] mutant. Time course of the release of pre-bound [3H]GDP from purified, recombinant ARL3 in the absence (ARL3) or presence of mouse wild type ARL13B or ARL13B[V358A] (mARL13B) are shown. See Methods for details. Error bars ± standard deviation.

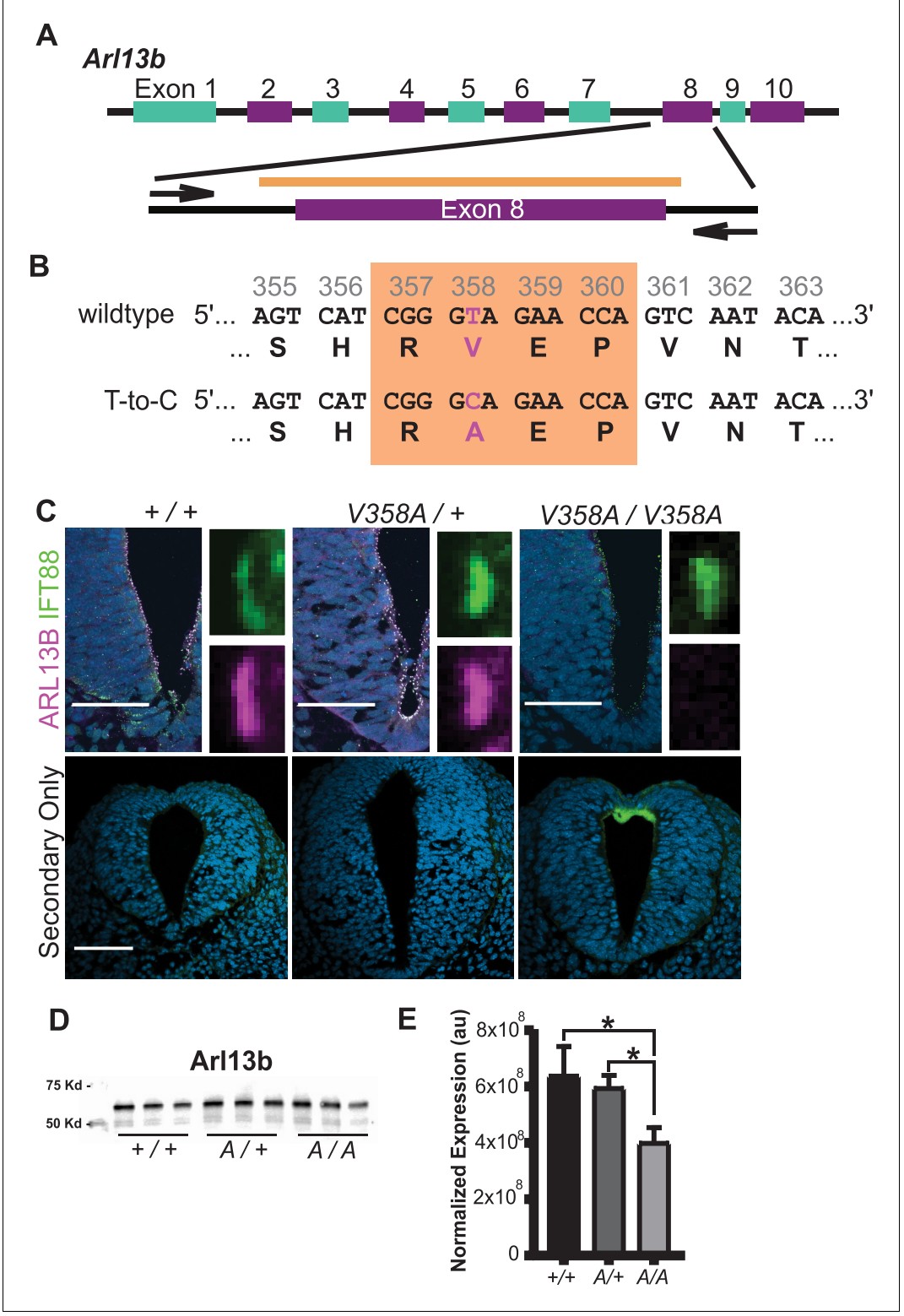

**Figure 2.** Generation of the *Arl13b*^V358A/V358A^ mouse. (**A**) Schematic of *Arl13b* gene and donor oligo (orange bar) at exon 8 used to generate the V358A causing point mutation. Arrows indicate primers used for allele validation. (**B**) *Arl13b* DNA and relevant amino acid sequence with the RVEP sequence in the orange box and the T-to-C mutation highlighted in pink. (**C**) Confocal images of cilia marker IFT88 (green) and ARL13B (magenta; NeuroMab) staining in neural tube of E10.5 somite-matched embryos. ARL13B-positive cilia are visible in *Arl13b*^+/+^ and *Arl13b*^V358A/+^, but not in *Arl1b*^V358A/V358A^ embryos. (See *Figure 2—figure supplement 1* for images of neural tube

*Figure 2 continued on next page*

*Figure 2 continued*

cilia under saturating conditions.) At least 5 embryos per genotype across five litters were examined. Scale bars are 50 µm. (D) ARL13B western blot of E10.5 whole embryo lysates, wild type (+/+), *Arl13b*$^{V358A/+}$ (A/+) and *Arl13b*$^{V358A/V358A}$ (A/A) and (E) quantification presented as average intensity normalized to total protein ± standard deviation. Representative blot of whole embryo lysates (n = 3 embryos per genotype with technical duplicate of each). *p<0.05, one-way ANOVA and Tukey's multiple comparison.

The online version of this article includes the following source data and figure supplement(s) for figure 2:

**Source data 1.** ARL13B western blot quantification.
**Figure supplement 1.** Overexposure of *Arl13b*$^{V358A/V358A}$ cilia in E10.5 neural tube reveals no clear ARL13B$^{V358A}$ presence in cilia.
**Figure supplement 2.** Overexposure of *Arl13b*$^{V358A/V358A}$ cilia in MEFs reveals no clear ARL13B$^{V358A}$ presence in cilia.

consistent with this point mutation having no effect on ARL13B GEF function (*Figure 1*). This result is consistent with GEF activity being conserved within the protein's GTPase domain while the V358A mutation is located in the C-terminal domain (*Gotthardt et al., 2015*). These data indicate that ARL13B$^{V358A}$ retains all known ARL13B biochemical activities, suggesting that the V358A mutation only disrupts ARL13B localization.

## CRISPR engineered *Arl13b*$^{V358A/V358A}$ mice express cilia-excluded ARL13B protein

To determine the consequences of ARL13B$^{V358A}$ expression *in vivo*, we used CRISPR/Cas9 editing to change residue 358 from valine to alanine (*Figure 2A*). We performed sequencing of the region that flanked exon 8, using primers outside the region of the donor oligonucleotide. We confirmed the heterozygous T to C base pair change for a valine to alanine change at residue 358 (*Figure 2B*). We backcrossed heterozygous F1 progeny to FVB/NJ for three generations before analysis to minimize any off target confounds (see details in Methods).

To determine whether ARL13B$^{V358A}$ was detectable in neural tube cilia, we performed immunofluorescence using antibodies directed against ARL13B and the cilia marker IFT88. At embryonic day (E)10.5, we saw IFT88 staining, indicating the presence of cilia. In neural tube cilia of *Arl13b*$^{+/+}$ and *Arl13b*$^{V358A/+}$ embryos, we observed equivalent ARL13B staining but could not see ARL13B signal in neural tube cilia of *Arl13b*$^{V358A/V358A}$ embryos (*Figure 2C*). We exposed the images longer, to saturation, and while we observed low-level staining throughout the cell, we saw no ciliary ARL13B indicating the ARL13B$^{V358A}$ protein is undetectable in cilia *in vivo* (*Figure 2—figure supplement 1*).

One possible explanation for the absence of ciliary ARL13B in the *Arl13b*$^{V358A/V358A}$ embryos is that the ARL13B$^{V358A}$ protein is not expressed or is unstable. To determine whether ARL13B protein expression was affected by the V358A mutation, we performed western blots on E10.5 whole embryo lysates (*Figure 2D*). We found a ~ 7% (±20%) reduction of ARL13B levels in *Arl13b*$^{V358A/+}$ embryos and a ~ 37% (±6%) decrease of ARL13B levels in *Arl13b*$^{V358A/V358A}$ embryos, compared to WT (*Figure 2E*). This decrease may reflect a change in ARL13B$^{V358A}$ stability compared to ciliary ARL13B or could signify ARL13B$^{V358A}$ protein having a distinct half-life when localizing to different cellular compartments. Regardless, these data indicate that ARL13B$^{V358A}$ is expressed in *Arl13b*$^{V358A/V358A}$ embryos.

## ARL13B$^{V358A}$ protein is undetectable in cilia in mouse embryonic fibroblasts

To more closely investigate whether any ARL13B$^{V358A}$ protein could localize to cilia, we generated immortalized mouse embryonic fibroblasts (MEFs). We performed double labeling to first identify cilia using acetylated α-tubulin or IFT88 antibodies and subsequently determined whether ARL13B is present in cilia. We used five distinct antibodies against ARL13B: four antibodies against amino acids 208–427 of the mouse ARL13B protein (one mouse monoclonal (NeuroMab), three rabbit polyclonal antibodies *Caspary et al., 2007*), as well as one antibody against the full-length human ARL13B protein (rabbit polyclonal (Protein Tech)). By eye, we observed strong ciliary staining of ARL13B with each of the antibodies in *Arl13b*$^{+/+}$ and *Arl13b*$^{V358A/+}$ cells, but we were unable to identify any evidence of ARL13B staining above background within cilia in *Arl13b*$^{V358A/V358A}$ cells with any of these

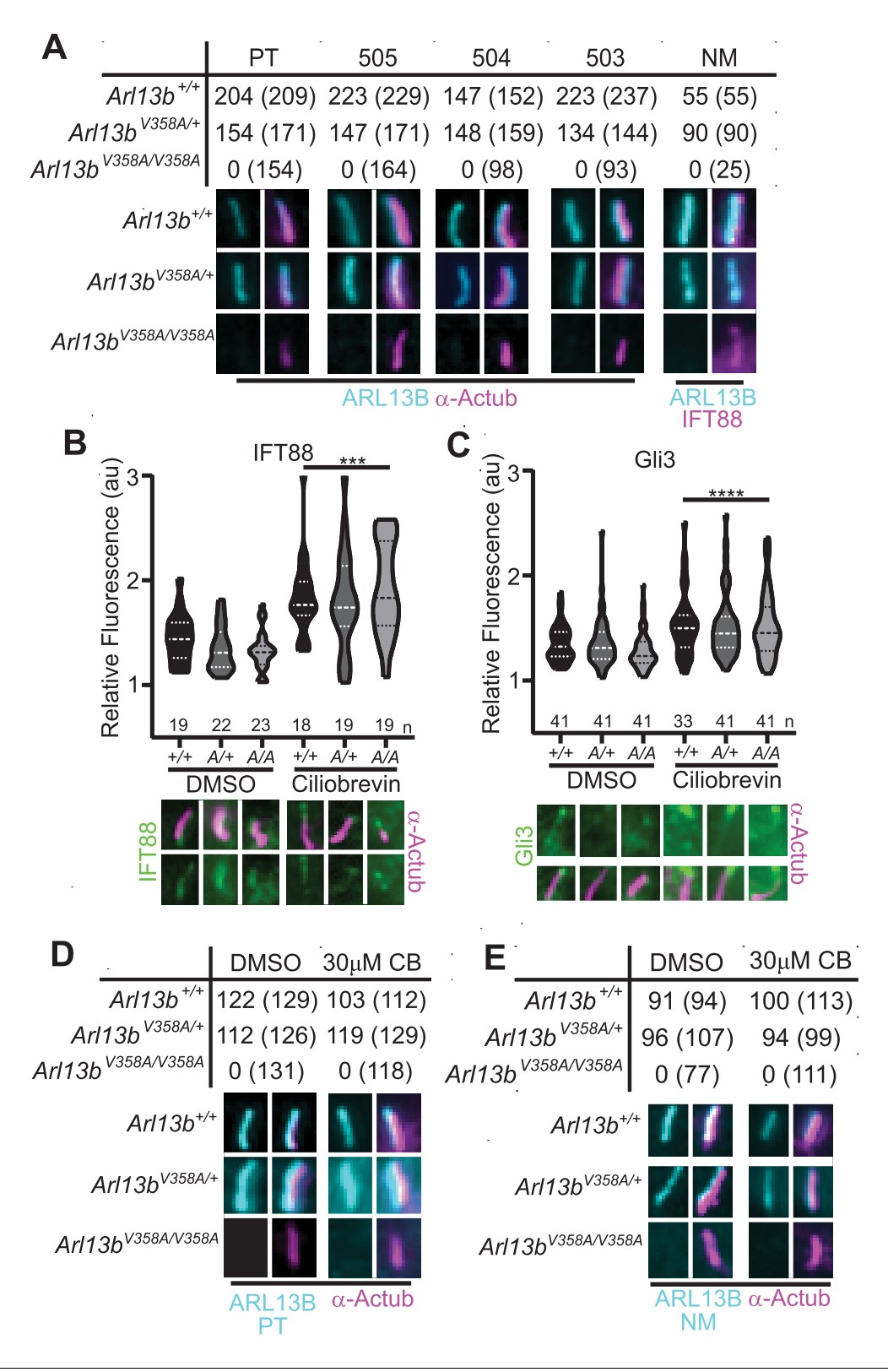

**Figure 3.** ARL13B[V358A] is undetectable in cilia and cannot be enriched by inhibition of retrograde transport. (**A**) Antibodies against ciliary markers acetylated α-tubulin or IFT88 (magenta) and ARL13B (cyan) in *Arl13b*[V358A/V358A] MEFs. Representative images show staining for five indicated ARL13B antibodies: (PT) polyclonal rabbit antibody
*Figure 3 continued on next page*

*Figure 3 continued*

against full-length human ARL13B from ProteinTech, (503, 504, 505) polyclonal rabbit sera from three distinct rabbits raised against C-terminus of mouse ARL13B (amino acids 208–427) (*Caspary et al., 2007*), and (NM) monoclonal mouse antibody against C terminus of mouse ARL13B from NeuroMab. *Arl13b$^{+/+}$* and *Arl13b$^{V358A/+}$* show ciliary ARL13B staining. Table lists ARL13B-positive cilia and the total number of cilia identified by acetylated α-tubulin or IFT88 antibody in parentheses. Cilia appear shorter in *Arl13b$^{V358A/V358A}$* cells (see *Figure 6*). (B) IFT88 and (C) Gli3 (green) is enriched in the tips of cilia in *Arl13b$^{+/+}$* (+/+), *Arl13b$^{V358A/+}$* (A/+) and *Arl13b$^{V358A/V358A}$* (A/A) cells following ciliobrevin-D treatment. Violin plots depict relative fluorescence of IFT88 and Gli3 at cilia tip to cell body with number of cilia measured (n) listed beneath each plot. (D and E) Table lists ARL13B (cyan) positive cilia (rabbit anti-ARL13B; ProteinTech or mouse anti-ARL13B; NeuroMab) and the total number of cilia (acetylated α-tubulin: magenta) examined in control (DMSO) and ciliobrevin-D treated (30 µM CB) cell lines. Representative images show staining for cilia and ARL13B. Staining of IFT88 and Gli3 analyzed by two-way ANOVA and Sidak's multiple comparisons. (***p<0.001, ****p<0.0001).

The online version of this article includes the following source data and figure supplement(s) for figure 3:

**Source data 1.** Analysis of ARL13B in MEFs.
**Figure supplement 1.** Endogenous ARL13B is undetectable in the cell body of cilia mutant *Ift172$^{wim/wim}$* cells.

---

five antibodies (*Figure 3A*). Taken together, these data support our conclusion that ARL13B$^{V358A}$ is not detectable in cilia using the currently available, validated ARL13B antibodies. While we observed no evidence of detectable ciliary ARL13B in *Arl13b$^{V358A/V358A}$* MEFs, it is possible that a small amount of ARL13B$^{V358A}$ is constantly trafficking in and out of cilia, at steady-state levels that remain below the limits of detection. To begin to address this possibility, we blocked retrograde ciliary transport, reasoning that any ARL13B undergoing trafficking in and out of cilia would accumulate. We treated cells with ciliobrevin-D, which blocks the retrograde motor protein dynein (*Firestone et al., 2012*). As a positive control, we examined IFT88 and Gli3, a ciliary protein and a Hh component, respectively; both are known to accumulate at ciliary tips upon blocking of retrograde transport. We found IFT88 and Gli3 enrichment at ciliary tips in *Arl13b$^{+/+}$*, *Arl13b$^{V358A/+}$* and *Arl13b$^{V358A/V358A}$* cells upon ciliobrevin-D treatment relative to the respective DMSO-treated controls with no difference in IFT88 or Gli3 staining among the three genotypes (*Figure 3B and C*). To determine whether ARL13B accumulated in cilia of ciliobrevin-D-treated MEFs, we examined cells co-stained for acetylated α-tubulin and ARL13B. In *Arl13b$^{+/+}$* and *Arl13b$^{V358A/+}$* cells, we saw that about 90% of acetylated α-tubulin-positive cilia also stained for ARL13B in DMSO-treated control or ciliobrevin-D treated MEFs (*Figure 3D and E*). In *Arl13b$^{V358A/V358A}$* MEFs, we saw no ciliary ARL13B staining in DMSO-treated control or ciliobrevin-D-treated cells using two antibodies against distinct epitopes (*Figure 3D and E*). Thus, even when retrograde traffic out of cilia is disrupted, we were unable to detect ARL13B protein in cilia in *Arl13b$^{V358A/V358A}$* MEFs.

We re-examined these data and repeated our analyses using over-exposed images. After defining the region of interest using the acetylated α-tubulin staining, we subsequently overexposed the ARL13B channel five-fold relative to the images in *Figure 3* and acquired measurements at the cilium and the cell body (*Figure 2—figure supplement 2*). As a control, we used *Arl13b$^{hnn/hnn}$* MEFs, which are devoid of any ARL13B, and obtained a ratio of 1.0, with or without ciliobrevin-D treatment. We found the same ratio when we analyzed *Arl13b$^{V358A/V358A}$* MEFs consistent with ARL13B being absent from the cilia. We observed a few instances of ARL13B appearing to co-localize with acetylated α-tubulin, but these were rare (<5%, 2/49) and occurred in <u>both</u> *Arl13b$^{V358A/V358A}$* and *Arl13b$^{hnn/hnn}$* (null) cells indicating this is the background staining of the primary antibody. We extended our analysis of overexposed images in *Arl13b$^{V358A/V358A}$* and *Arl13b$^{hnn/hnn}$* neural tube sections four-fold relative to the images in *Figure 2*; we identified no overlap of overexposed ARL13B with cilia marker IFT88 (*Figure 2—figure supplement 1*). While it is formally possible that an extremely small amount of ARL13B$^{V358A}$ remains in cilia, we are not able to find evidence of it; thus, we designate ARL13B$^{V358A}$ protein as cilia-excluded ARL13B.

We did not observe any obvious cellular ARL13B signal in the cells expressing cilia-excluded ARL13B so we investigated whether we could detect cellular ARL13B in cells lacking cilia, *Ift172$^{wim/wim}$* MEFs. As controls we used wildtype, *Arl13b$^{hnn/hnn}$* (lacking ARL13B), and *Ift172$^{wim/wim}$ Arl13b$^{hnn/hnn}$* (lacking ARL13B and cilia) MEFs. We only detected ARL13B staining in *Ift172$^{wim/wim}$* cells upon overexposure and found it was modestly detectable above the background level we observed in

*Arl13b*[hnn/hnn] or *Ift172*[wim/wim] *Arl13b*[hnn/hnn] cells (*Figure 3—figure supplement 1*). Thus, cellular ARL13B is expressed at an extremely low level.

## *Arl13b*[V358A/V358A] mice are viable and fertile

In order to determine the phenotypic consequence of the *Arl13b*[V358A] allele, we intercrossed *Arl13b*[V358A/+] mice. We observed progeny in Mendelian proportions with an average of 7.3 pups per litter, consistent with the reported FVB/NJ litter size of 7–9 (*Murray et al., 2010*; *Table 1*). To test whether homozygous mice breed normally, we crossed *Arl13b*[V358A/V358A] males to heterozygous or homozygous females. We found the *Arl13b*[V358A] allele segregated in Mendelian proportions and the litter sizes were normal indicating that *Arl13b*[V358A] does not impact viability, fertility or fecundity.

## ARL13B[V358A] permits normal embryonic development and Shh signaling

Loss of *Arl13b* leads to morphologically abnormal embryos, lethality and neural patterning defects (*Caspary et al., 2007*). As ARL13B[V358A] overcame the embryonic lethality, we examined overall embryo morphology at E9.5, E10.5 and E12.5. We found the overall morphology of *Arl13b*[V358A/V358A] embryos resembled those of *Arl13b*[+/+] and *Arl13b*[V358A/+] embryos indicating that ARL13B[V358A] did not lead to gross morphological defects. (*Figure 4A–I*).

At E9.5, Shh is normally expressed in the notochord as well as the floor plate of the ventral neural tube. Moving dorsally, the adjacent domains express Nkx2.2 and subsequently Olig2. In the *Arl13b*[hnn/hnn] (null) neural tube, the Shh-positive columnar cells of the floor plate are absent resulting in both ventral and dorsal expansion of intermediate Shh-dependent cell fates such as Olig2 (*Caspary et al., 2007*). To determine whether ARL13B[V358A] disrupts Shh signaling, we examined neural tube patterning in *Arl13b*[+/+], *Arl13b*[V358A/+], and *Arl13b*[V358A/V358A] embryos at E9.5, E10.5, and E12.5. At E9.5 we observed no differences in Shh, Nkx2.2 or Olig2 among *Arl13b*[+/+], *Arl13b*[V358A/+], and *Arl13b*[V358A/V358A] embryos (*Figure 4A–C*). Cell fates in the neural tube are specified by both the concentration and duration of Shh signaling so we examined neural patterning at subsequent stages (*Dessaud et al., 2007*). By E10.5, some Olig2 precursors have differentiated to HB9+ motor neurons and all Shh-responsive cells express Nkx6.1. We found that Olig2, HB9 and Nkx6.1 positive cells are normally distributed in all 3 genotypes at E10.5 and E12.5 (*Figure 4D–I*). These data indicate that ARL13B[V358A] mediates normal Shh signaling.

## ARL13B regulates ciliary enrichment of Shh components from outside cilia

In *Arl13b*[hnn/hnn] cells, Smo is enriched in cilia in a Shh-independent manner, which may be due to defective trafficking of Smo, as many ARL family members regulate protein trafficking (*Lim et al., 2011*; *Larkins et al., 2011*). To assess Smo enrichment in *Arl13b*[V358A/V358A] embryos, we stained for the cilia marker acetylated α-tubulin and Smo in E10.5 embryos. Smo appeared enriched normally in ventral floor plate cilia and the dorsal boundary of Smo ciliary enrichment in ventral neural

**Table 1.** Genotype of mice born to heterozygous and/or homozygous carrier parents. Data analyzed by chi-squared test.

| MICE | Sex | wildtype | V358A/+ | V358A/V358A | |
|---|---|---|---|---|---|
| Het × Het Avg. Litter 7.3 | M (46.5%) | 3 | 11 | 6 | |
| | F (53.5%) | 6 | 8 | 9 | χ2 = 0.60 |
| | % of Total | 20.9 | 44.2 | 34.9 | |
| Het × Hom Avg. Litter 7.5 | M (60%) | - | 10 | 11 | |
| | F (40%) | - | 6 | 8 | χ2 = 0.69 |
| | % of Total | - | 45.7 | 54.3 | |
| Hom × Hom Avg. Litter 9.5 | M (47.4%) | - | - | 9 | |
| | F (52.6%) | - | - | 10 | |

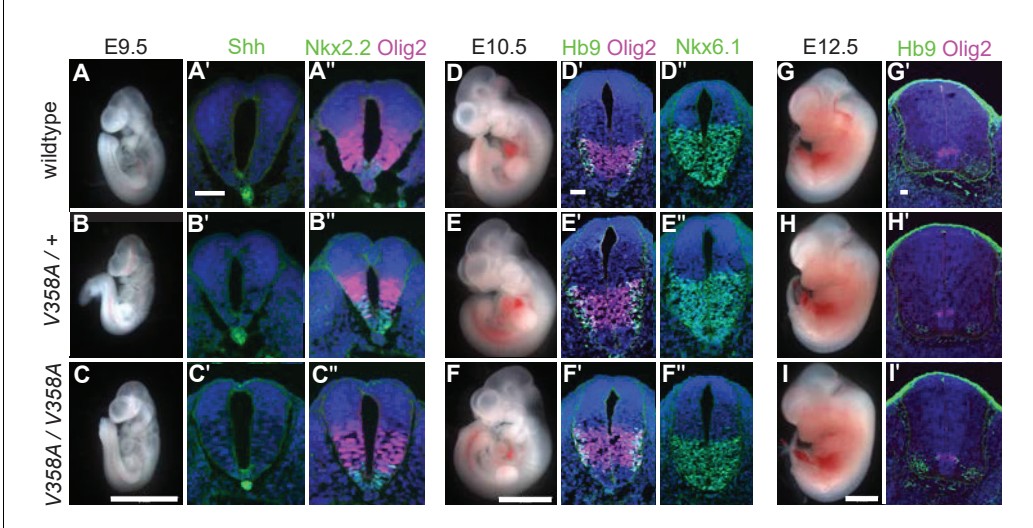

**Figure 4.** ARL13B^V358A mediates normal Shh signaling and neural tube patterning. (**A–I**) Whole embryo and neural tube sections of somite-matched littermates at E9.5, E10.5, and E12.5 stained with indicated markers of cell fate. Whole embryo scale bars: 2 mm. (A'-I' and A'-F') Shh-dependent neural tube patterning at three separate time points. Cell fate markers are listed above each image. All neural tube scale bars are 50 μm. Neural tube patterning was examined in five embryos for each embryonic stage. E, embryonic day.

progenitors was identical in *Arl13b*^+/+, *Arl13b*^V358A/+, and *Arl13b*^V358A/V358A samples indicating ARL13B^V358A mediates normal ciliary Smo enrichment (*Figure 5A*).

To examine Smo traffic in response to Shh stimulation, we treated MEFs with 0.5% FBS or Shh-conditioned media for 24 hours and stained for Smo. As expected in control *Arl13b*^hnn/hnn cells, we saw ciliary Smo in unstimulated MEFs which increased upon stimulation with Shh-conditioned media (*Larkins et al., 2011*; *Figure 5B*). In *Arl13b*^+/+, *Arl13b*^V358A/+, and *Arl13b*^V358A/V358A MEF cilia, we saw ciliary enrichment of Smo upon Shh stimulation over their respective unstimulated controls (*Figure 5B*, left). We found no difference in Smo enrichment among these cell lines (*Figure 5B*, right). Thus, despite ARL13B being critical for Shh-dependent Smo ciliary enrichment, ARL13B^V358A regulates Smo localization normally, arguing this function of ARL13B can occur when the protein is not in cilia.

In addition to aberrations in Smo trafficking, loss of *Arl13b* leads to changes in the cilia localization of other Shh components in MEFs (*Larkins et al., 2011*). To determine whether these components are disrupted by ARL13B^V358A, we examined Gli2, Gli3, Sufu and Ptch1 in MEFs. We observed no differences in distribution of Gli2 or Ptch1 among any of the examined genotypes (*Figure 5C,F*). In contrast, as we previously reported, we observed more Gli3 at the ciliary tip of *Arl13b*^hnn/hnn cells and increased Sufu in *Arl13b*^hnn/hnn cilia compared to wild type controls (*Larkins et al., 2011*; *Figure 5D,E*). Thus, Shh components are normally localized in *Arl13b*^V358A/V358A MEFs consistent with the normal Shh signal transduction observed in *Arl13b*^V358A/V358A embryos (*Figure 4*).

## ARL13B regulates ciliary enrichment of ARL3 and INPP5E from within cilia

We next examined the role of ARL13B^V358A in the localization of other ciliary proteins. ARL13B is the GEF for ARL3, and we showed that ARL13B^V358A GEF activity for ARL3 is retained (*Figure 1*). ARL13B is also essential for cilia localization of INPP5E, as the proteins are in a common complex (*Humbert et al., 2012*). To determine whether the ciliary localization of either ARL3 or INPP5E was affected by ARL13B^V358A, we performed immunofluorescence on MEFs. Ciliary ARL3 staining appeared the same in *Arl13b*^+/+ and *Arl13b*^V358A/+ MEFs, however ARL3 was not detectable in *Arl13b*^V358A/V358A or *Arl13b*^hnn/hnn cilia (*Figure 5G*). INPP5E was normally detectable in *Arl13b*^+/+ and *Arl13b*^V358A/+ MEF cilia, but not in *Arl13b*^V358A/V358A and *Arl13b*^hnn/hnn MEF cilia (*Figure 5H*).

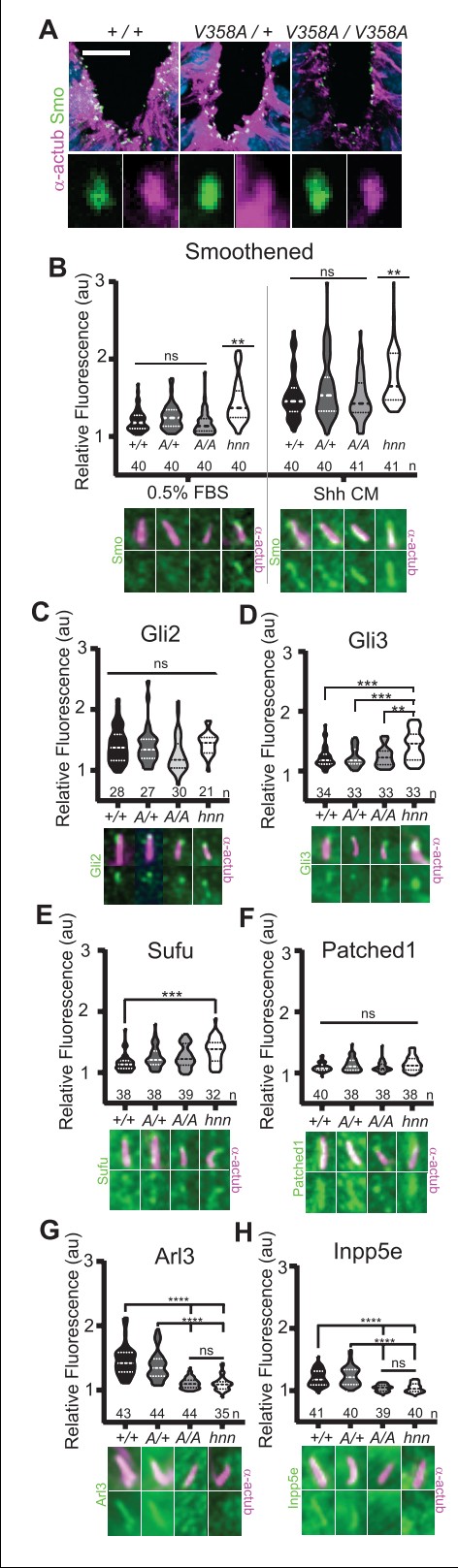

**Figure 5.** ARL13B[V358A] mediates normal ciliary enrichment of Shh components, but not ARL3 or INPP5E. (**A**) Smo (green) enrichment in ventral neural

*Figure 5 continued on next page*

These results indicate that ARL13B is required in cilia for the proper localization of ARL3 and INPP5E to the cilium.

## Ciliary ARL13B is required for normal cell ciliation and ciliary length

Loss of *Arl13b* leads to defects in the percentage of ciliated MEFs and in cilia length (*Caspary et al., 2007*; *Larkins et al., 2011*). To test whether ARL13B[V358A] impacted cell ciliation, we examined immortalized MEFs and counted the number of ciliated cells 24 hours after induction of ciliation by growth in low serum (0.5% FBS). We found 73% (±11%) of *Arl13b[+/+]* and 75%, (±9.1%) of *Arl13b[V358A/+]* MEFs are ciliated, consistent with published results. In contrast, we found 45% (±8.7%) of *Arl13b[V358A/V358A]* cells and 53% (±6.4%) of *Arl13b[hnn/hnn]* cells had cilia (*Figure 6A*). Thus, *Arl13b[V358A/V358A]* cells exhibit a similar deficit in percentage of ciliated MEFs as complete loss of ARL13B function.

*Arl13b[hnn/hnn]* cilia are about half as long as wild type in both embryos and MEFs (*Caspary et al., 2007*; *Larkins et al., 2011*). To determine whether cilia length was affected by ARL13B[V358A], we stained MEFs with acetylated α−tubulin and measured the length of the axoneme. We found the average cilia length in *Arl13b[+/+]* and Arl13b[V358A/+] MEFs was similar, 2.7 ± 0.8 μm and 2.6 ± 0.9 μm, respectively. However, we found *Arl13b[V358A/V358A]* MEF cilia were shorter, 1.9 ± 0.7 μm, similar to *Arl13b[hnn/hnn]* MEF cilia which were 2.15 ± 0.8 μm (*Figure 6B*). These results indicate that ARL13B[V358A] phenocopies complete loss of ARL13B for ciliation and cilia length, indicating these ARL13B functions require ARL13B in cilia. Furthermore, these data show that ARL13B function within cilia is distinct from ARL13B function outside of cilia in a subset of activities, and that the cilia defects and Shh defects in the complete absence of ARL13B can be uncoupled.

## Discussion

Here we show that ARL13B's role(s) in cell ciliation and cilia length, along with the ciliary enrichment of a subset of proteins, can be uncoupled from ARL13B's function in regulating Shh signal transduction (*Figure 7*). Furthermore, we demonstrate this functional distinction correlates with ARL13B spatial localization to the cilium. By generating a mouse expressing a cilia-excluded variant of ARL13B from the endogenous locus, we showed ARL13B[V358A] protein is not detectable in

*Figure 5 continued*

tube cilia (acetylated α-tubulin: magenta) is normal in E10.5 embryos. Images are confocal projections. Scale bar is 25 μm. (B-H, Top) Quantification of average fluorescence intensity in the tip of the cilium (Gli2, Gli3, and Sufu) or the entire cilium (Ptch1, Smo, ARL3, and INPP5E) relative to background level. Violin plots depict relative fluorescent intensity per cilium with number of cilia examined below each plot. (B-H, Bottom) Representative images for each condition and cell type with the cilia marker acetylated α-tubulin (magenta) and indicated protein (green). *Arl13b^{+/+}* (+/+), *Arl13b^{V358A/+}* (A/+), *Arl13b^{V358A/V358A}* (A/A) and *Arl13b^{hnn/hnn}* (hnn). Data analyzed by one-way ANOVA and Tukey's multiple comparisons, except Smo data analyzed by two-way ANOVA and 16 comparisons, corrected $p<0.003$ deemed significant. (**$p<0.01$, ***$p<0.001$, ****$p<0.0001$).

The online version of this article includes the following source data for figure 5:

**Source data 1.** Immunofluorescence of cilia proteins.

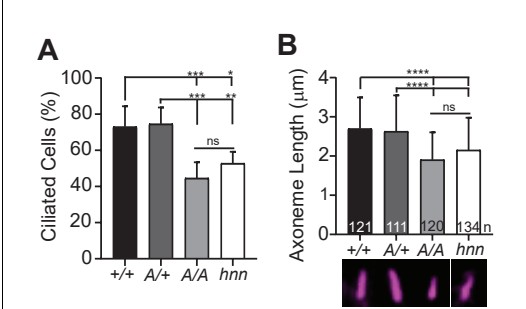

**Figure 6.** ARL13B^{V358A} results in decreased ciliation rates and short cilia. (**A**) Quantification of ciliation rates in all cell types; *Arl13b^{+/+}* (+/+), *Arl13b^{V358A/+}* (A/+), *Arl13b^{V358A/V358A}* (A/A) and *Arl13b^{hnn/hnn}* (hnn). Fewer *Arl13b^{V358A/V358A}* and *Arl13b^{hnn/hnn}* MEFs form cilia compared to *Arl13b^{+/+}* or *Arl13b^{V358A/+}* cells. (**B**) Quantification of axoneme length as labeled by acetylated α-tubulin (magenta) in indicated MEFs. Data are presented as mean (μm)± standard deviation with the number of cilia measured per genotype depicted at the base of each bar. Data analyzed by one-way ANOVA and Tukey's multiple comparisons. (*$p<0.05$, **$p<0.01$, ***$p<0.001$, ****$p<0.0001$).

The online version of this article includes the following source data for figure 6:

**Source data 1.** Analysis of ciliated MEFs.

cilia of embryonic neural tube or MEFs, even upon retrograde ciliary dynein traffic blockade. Furthermore, we detected 30% less ARL13B protein overall in *Arl13b^{V358A/V358A}* embryos compared to control embryos. While this reduction is statistically significant, it is unclear whether it is biologically significant given that *Arl13b^{hnn/hnn}* null mutations are recessive and no *Arl13b^{hnn/+}* heterozygous phenotype is reported (*Caspary et al., 2007*; *Larkins et al., 2011*).

In contrast to the E13.5 lethality of *Arl13b^{hnn/hnn}* embryos, we found *Arl13b^{V358A/V358A}* mice were viable and fertile with correct patterning of the neural tube, indicating normal Shh signal transduction. These results are consistent with our data showing that ARL13B^{V358A} protein retains all known biochemical function including GEF activity (*Mariani et al., 2016*; *Ivanova et al., 2017*). However, we observed several *Arl13b^{hnn/hnn}* phenotypes in *Arl13b^{V358A/V358A}* cells, including loss of ARL3 and INPP5E ciliary enrichment, along with low percentage of ciliated cells and shorter average cilia length. Taken together, our data show that despite the normal high ciliary enrichment of wild type ARL13B and the requirement of cilia for Shh signaling, cilia-excluded ARL13B^{V358A} is sufficient for Shh signal transduction.

We regard ARL13B^{V358A} as cilia-excluded based on several lines of evidence. We did not detect ciliary ARL13B^{V358A} *in vivo* or *in vitro* using any of five validated ARL13B antibodies. These antibodies are against two antigens, the full-length protein or residues 208–427, and were independently generated so are likely to recognize a number of epitopes. Given that four of five antibodies are polyclonal and ARL13B^{V358A} displays normal GTP binding, intrinsic and GAP-stimulated GTP hydrolysis, and ARL3 GEF activity, it is likely that the ARL13B^{V358A} protein retains the wild type structure enabling antibody recognition. Indeed, we observed little or no loss in sensitivity in detecting protein in the immunoblot assays. Furthermore, we could not forcibly enrich ciliary ARL13B^{V358A} through retrograde transport blockade with ciliobrevin-D nor could we detect any ciliary enrichment of ARL13B^{V358A} upon overexposure of the relevant fluorescent channel. Alternative ways of trafficking proteins out of cilia include BBSome-dependent or diffusion-dependent mechanisms raising the possibility that ARL13B is trafficked via dynein-independent mechanisms. The fact that we observed ciliary phenotypes, namely short cilia and abnormal ciliary ARL3 and INPP5E localization in ARL13B^{V358A} expressing cells, argue that ARL13B normally performs these functions from within cilia (*Caspary et al., 2007*; *Larkins et al., 2011*; *Zhang et al., 2016*; *Humbert et al., 2012*; *Nozaki et al., 2017*). While we cannot exclude the possibility that sub-detectable levels of

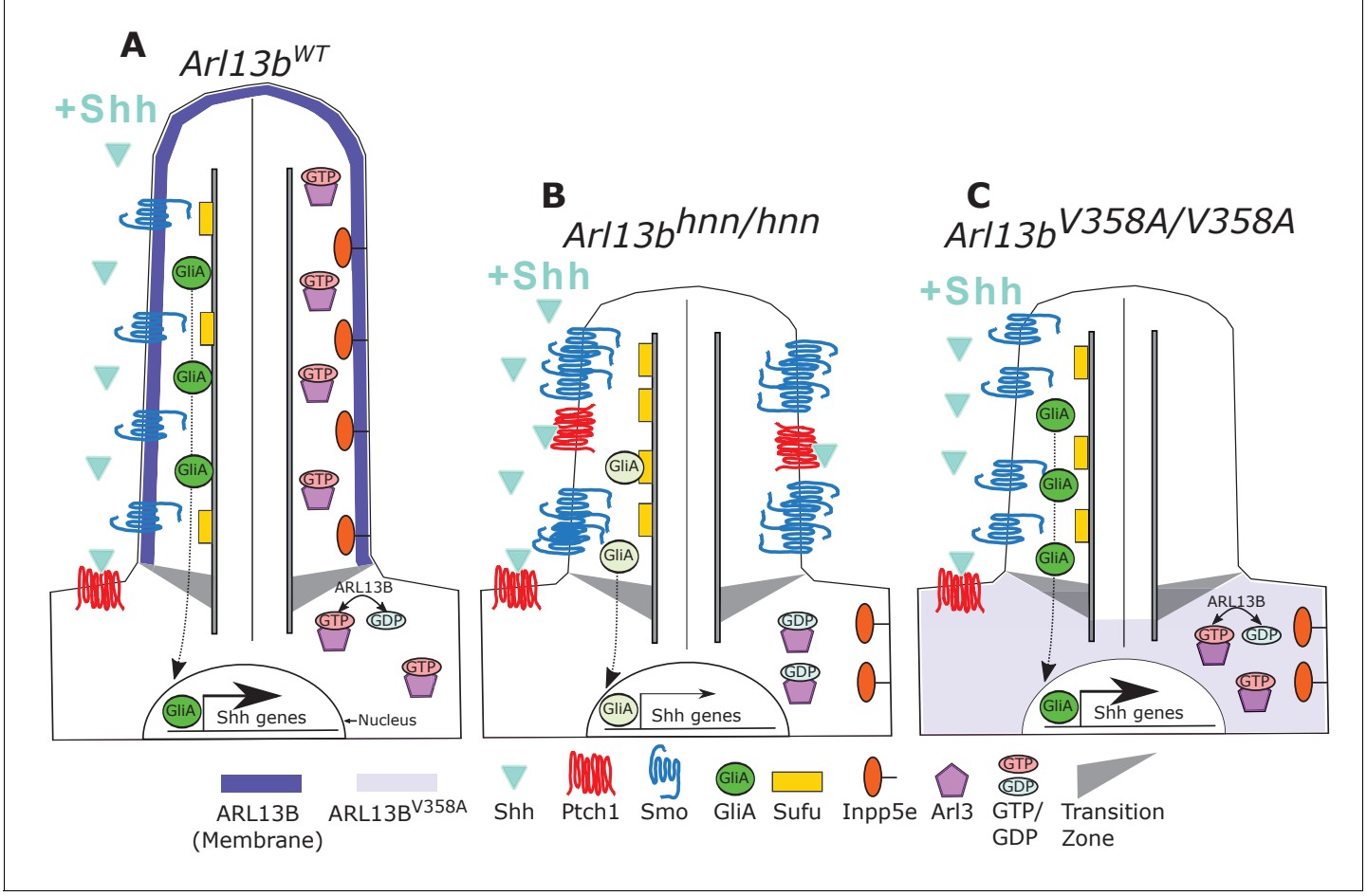

**Figure 7.** Model comparing complete loss of ARL13B function to ciliary loss of ARL13B function. Wildtype (left), *Arl13b^hnn/hnn* (middle), and *Arl13b^V358A/V358A* (right) cilia represented as two halves. On the left half is the organization of Shh components in the presence of Shh ligand and on the right half is the organization of ARL13B interactors ARL3 and INPP5E. (**A**) ARL13B associates with the ciliary membrane. In the presence of Shh, Ptch1 is removed from cilia and Smo is visibly enriched in cilia. Smo is activated which promotes the processing of full-length Gli transcription factors into their activator forms (GliA), that are shuttled out of the cilium to promote Shh-dependent gene transcription. In addition, cilia proteins ARL3 and INPP5E are localized to the primary cilium. (**B**) In *Arl13b^hnn/hnn* cells which are null for ARL13B, cilia are shorter than normal. Ciliary Ptch1 and Smo are visible, although Smo appears punctate instead of diffuse. In addition, loss of ARL13B decreases transcription of Shh-dependent genes due to lowered GliA. *Arl13b^hnn/hnn* cilia also display a failure of INPP5E and ARL3 to localize to the cilium. Because ARL13B is the GEF for ARL3, we speculate in this schematic that ARL3 remains GDP bound in *Arl13b^hnn/hnn* cells. (**C**) In *Arl13b^V358A/V358A* cells, ARL13B^V358A is not detectable in cilia and appears diffuse within the cell body. *Arl13b^V358A/V358A* cilia, like *Arl13b^hnn/hnn* cilia, are shorter than wildtype. We observe normal Shh-dependent ciliary Smo enrichment and normal Shh transcriptional output. However, ARL3 and INPP5E are absent from cilia, indicating that ciliary ARL13B is required for ciliary residence of these proteins.

ARL13B^V358A are present and functional in cilia, we note such a level would need to be sufficient for Shh signaling yet insufficient for ARL3 or INPP5E ciliary localization along with proper regulation of cilia length.

Our data support ARL13B regulating different biological processes from its distinct subcellular localizations consistent with how other GTPases act from multiple sites in cells through different effectors. ARL13B^V358A disrupts cilia localization of INPP5E and ARL3, but not Shh components and thus provides a novel tool with which to identify additional ARL13B effectors. Such effectors may also inform us as to what subcellular compartment ARL13B functions from as the cellular ARL13B antibody staining we observe is inconclusive. It is surprising that ARL13B^V358A is sufficient to regulate Shh signaling because cilia are required for Shh signal transduction and because ARL13B is highly enriched in cilia (*Caspary et al., 2007*; *Huangfu et al., 2003*). These observations suggested that the ciliary defects observed in *Arl13b^hnn/hnn* mutants caused the Shh defects. However, our data

indicate that ARL13B regulates cilia length and Shh signaling through distinct localization. Thus, the cilia defects in *Arl13b^{hnn/hnn}* mutants do not underlie Shh misregulation.

Based on the normal cilia trafficking of Shh components in *Arl13b^{V358A/V358A}* mutants and the abnormal cilia trafficking of Shh components in the complete absence of ARL13B (*Larkins et al., 2011*), ARL13B likely regulates Shh signaling from the cell body by controlling Shh component traffic to and/or from the primary cilium. Both Smo and ARL13B traffic is linked to endosomes providing one possible non-ciliary organelle (*Barral et al., 2012*; *Wang et al., 2009*; *Milenkovic et al., 2009*). *Arl13b^{hnn/hnn}* cells display constitutive Smo ciliary enrichment along with little enrichment of Gli proteins at the ciliary tip (*Larkins et al., 2011*). Our data do not distinguish between ARL13B playing direct roles in traffic of multiple Shh components or whether the normal Smo ciliary enrichment with ARL13B^{V358A} subsequently causes normal cilia traffic of downstream components. ARL13B regulates a step downstream of activated Smo that controls transcription factor Gli activation, but not repression (*Caspary et al., 2007*; *Bay et al., 2018*). We argue from our results that this step is also intact in the presence of ARL13B^{V358A}.

The fact that ARL13B^{V358A} can mediate normal ciliary Smo enrichment is especially interesting given that ARL3 and INPP5E localization to cilia is compromised by this mutation. This suggests that ARL13B controls cilia enrichment via multiple localizations and effectors. This is consistent with our understanding of ARF family members, as they are known to perform multiple tasks from different sites within a cell (*Sztul et al., 2019*). We speculate that ARL3 residence in cilia may require that ARL3 be in (or at least cycle through) its activated, GTP-bound conformation as ARL13B^{V358A} retains its ARL3 GEF activity, but not its cilia localization. ARL13B is in a common protein complex with INPP5E so absence of ARL13B from cilia may disrupt formation or maintenance of the complex there (*Humbert et al., 2012*; *Nozaki et al., 2017*). INPP5E regulates Shh signaling through regulation of the phosphoinositol composition of the ciliary membrane. INPP5E loss results in increased ciliary $PIP_2$ and enrichment of Shh repressors in cilia thus resulting in lowered Shh response (*Garcia-Gonzalo et al., 2015*; *Chávez et al., 2015*; *Dyson et al., 2017*). It is not clear why the absence of ciliary INPP5E in ARL13B^{V358A} cells does not lead to aberrant Shh signal transduction.

In *Arl13b^{hnn/hnn}* embryos and MEFs, cilia are shorter than normal and have a microtubule defect where the B-tubule of the microtubule outer doublet is open (*Caspary et al., 2007*; *Larkins et al., 2011*). Consistent with this, we observed shorter cilia in *Arl13b^{V358A/V358A}* MEFs. Similarly, we observed a comparable reduction in the percent of cilia in *Arl13b^{hnn/hnn}* and *Arl13b^{V358A/V358A}* MEFs compared to wild type MEFs. While loss of ARL13B results in short cilia, increased expression of ARL13B promotes cilia elongation so it is not yet clear what ARL13B's role is in controlling cilia length (*Lu et al., 2015*; *Hori et al., 2008*; *Caspary et al., 2007*; *Larkins et al., 2011*). Additionally, it is unclear whether the mechanism from within cilia through which ARL13B controls length or percent of ciliated cells is the same, or distinct from, the mechanism through which ARL13B regulates ARL3 or INPP5E residence in cilia.

Perhaps the most intriguing implication of our data pertains to understanding the evolution of cilia and Hh signaling. ARL13 functions in cilia formation and maintenance in *Chlamydomonas* and *C. elegans*, neither of which have Hh signaling, consistent with the ancient role of ARL13 controlling ciliation and cilia length (*Cevik et al., 2010*; *Cevik et al., 2013*; *Stolc et al., 2005*; *Miertzschke et al., 2014*). Our data support ARL13B retaining ARL13's ancient role in ciliogenesis. However, our data show that ARL13B does not work from within the cilium to regulate Shh component ciliary traffic or Shh signal transduction. As an ARL protein involved in membrane traffic, we speculate that ARL13B may have linked Shh component trafficking to the primary cilium, albeit from outside the cilium. That the ARL13 duplication could relate to the mechanism linking primary cilia and Hh signaling within the deuterostome lineage is a possibility worth exploring.

## Materials and methods

**Key resources table**

| Reagent type (species) or resource | Designation | Source or reference | Identifiers | Additional information |
|---|---|---|---|---|

*Continued on next page*

*Continued*

| Reagent type (species) or resource | Designation | Source or reference | Identifiers | Additional information |
|---|---|---|---|---|
| Gene (*M. musculus*) | *Arl13b* | | MGI Cat# 6115585, RRID:MGI:6115585 | |
| Genetic reagent (*M. musculus*) | *hennin* | PMID:17488627 | MGI Cat# 3580673, RRID:MGI:3580673 | *Arl13b* null allele |
| Genetic reagent (*M. musculus*) | *em1Tc (V358A)* | This paper | MGI: 6256969 | New CRISPR Point mutant |
| Genetic reagent (*M. musculus*) | FBV/NJ | Jackson Laboratory | Stock #001800 MGI:2163709 | |
| Cell lines (*M. musculus*) | Fibroblast (normal, embryonic) | This paper | | Maintained in Caspary lab |
| Antibody | Anti-Shh (Mouse Monoclonal) | Developmental Studies Hybridoma Bank | DSHB Cat# 5E1, RRID:AB_528466 | 1:5 |
| Antibody | Anti-Nkx2.2 (Mouse Monoclonal) | Developmental Studies Hybridoma Bank | DSHB Cat# 74.5A5, RRID:AB_531794 | 1:5 |
| Antibody | Anti-Hb9 (Mouse Monoclonal) | Developmental Studies Hybridoma Bank | DSHB Cat# 81.5C10, RRID:AB_2145209 | 1:5 |
| Antibody | Anti-Nkx6.1 (Mouse Monoclonal) | Developmental Studies Hybridoma Bank | DSHB Cat# F55A10, RRID:AB_532378 | 1:50 |
| Antibody | Anti-acetylated a-tubulin (Mouse Monoclonal) | Millipore Sigma | Sigma-Aldrich Cat# T6793, RRID:AB_477585 | 1:2500 |
| Antibody | Anti-Olig2 (Rabbit Polyclonal) | Millipore Sigma | Millipore Cat# AB9610, RRID:AB_570666 | 1:300 |
| Antibody | Anti-Arl13b (Mouse Monoclonal) | NeuroMab | UC Davis/NIH NeuroMab Facility Cat# 73–287, RRID:AB_11000053 | 1:1000 |
| Antibody | Anti-Arl13b (Rabbit Polyclonal | Protein Tech | Proteintech Cat# 17711–1-AP, RRID:AB_2060867 | 1:1000 |
| Antibody | Anti-Arl13b (Rabbit Polyclonal | PMID:17488627 | 503 | 1:1000 |
| Antibody | Anti-Arl13b (Rabbit Polyclonal | PMID:17488627 | 504 | 1:1000 |
| Antibody | Anti-Arl13b (Rabbit Polyclonal | PMID:17488627 | 505 | 1:1000 |
| Antibody | Anti-Smo (Rabbit Polyclonal) | K. Anderson | | 1:1000 |
| Antibody | Anti-IFT88 (Rabbit Polyclonal) | B. Yoder | | 1:1000 |
| Antibody | Anti-Arl3 (Rabbit Polyclonal) | PMID:8034651 | | 1:1000 |
| Antibody | Anti-Inpp5e (Rabbit Polyclonal) | Protein Tech | Proteintech Cat# 17797–1-AP, RRID:AB_2167120 | 1:150 |

*Continued on next page*

*Continued*

| Reagent type (species) or resource | Designation | Source or reference | Identifiers | Additional information |
|---|---|---|---|---|
| Antibody | Anti-Gli2 (Guinea Pig Polyclonal) | J. Eggenschwiler | | 1:200 |
| Antibody | Anti-Gli3 (Goat Polyclonal) | R and D | R and D Systems Cat# AF3690, RRID:AB_2232499 | 1:200 |
| Antibody | Anti-Ptch1 (Rabbit Polyclonal) | R. Rohatgi | | 1:150 |
| Antibody | Anti-Sufu (Goat Polyclonal) | Santa Cruz | Santa Cruz Biotechnology Cat# sc-10933, RRID:AB_671172 | 1:100 |
| Antibody | Alexa Fluor goat anti-mouse IgG2a 488 | ThermoFisher | Thermo Fisher Scientific Cat# A-21131, RRID:AB_2535771 | 1:300 |
| Antibody | Alexa Fluor goat anti-mouse IgG1 488 | ThermoFisher | Thermo Fisher Scientific Cat# A-21121, RRID:AB_2535764 | 1:300 |
| Antibody | Alexa Fluor goat anti-mouse Ig 488 | ThermoFisher | Molecular Probes Cat# A-11029, RRID:AB_138404 | 1:300 |
| Antibody | Alexa Fluor goat anti-mouse IgG 568 | ThermoFisher | Thermo Fisher Scientific Cat# A-11031, RRID:AB_144696 | 1:300 |
| Antibody | Alexa Fluor donkey anti-rabbit IgG 488 | ThermoFisher | Thermo Fisher Scientific Cat# A-21206, RRID:AB_2535792 | 1:300 |
| Antibody | Alexa Fluor donkey anti-rabbit IgG 555 | ThermoFisher | Thermo Fisher Scientific Cat# A-31572, RRID:AB_162543 | 1:300 |
| Antibody | Alexa Fluor goat anti-rabbit IgG 568 | ThermoFisher | Thermo Fisher Scientific Cat# A-11011, RRID:AB_143157 | 1:300 |
| Antibody | Alexa Fluor goat anti-mouse IgG2b 568 | ThermoFisher | Thermo Fisher Scientific Cat# A-21147, RRID:AB_2535783 | 1:300 |
| Antibody | Hoechst nuclear stain | Millipore Sigma | 94403 | 1:3000 |
| Sequence-based reagent | CRISPR gRNA | Millpore Sigma, this paper | | CCAGTCAATACAGA CGAGTCTA |
| Sequence-based reagent | CRISPR donor oligo | Millpore Sigma, this paper | | CCTATATTCTTCTAGAAAAC AGTAAGAAGAAAACCAAGAA ACTACGAATGAAAAGGAGTC ATCGGGCAGAACCAGTGAAT ACAGACGAGTCTACTCCAAA GAGTCCCACGCCTCCCCAAC |
| Sequence-based reagent | F-231-Cac8I | This Paper | | PCR Primer AAGAATGAAAAGGAGTCAGCG |
| Sequence-based reagent | REV-1 | This Paper | | PCR Primer TGAACCGCTAATGGGAAACT |

*Continued on next page*

*Continued*

| Reagent type (species) or resource | Designation | Source or reference | Identifiers | Additional information |
|---|---|---|---|---|
| Peptide, recombinant protein | Arl13b$^{WT}$-GST | This Paper | | Purified from cells |
| Peptide, recombinant protein | Arl13b$^{V358A}$-GST | This Paper | | Purified from cells |
| Peptide, recombinant protein | Arl3 (human) | PMID:11303027 | | Purified from cells |
| Peptide, recombinant protein | Cas9 | Millipore Sigma | C120010 | 50 µg |
| Peptide, recombinant protein | Cac8I | New England Biolabs | R0579L | Restriction enzyme |
| Chemical compound, drug | Ciliobrevin-D | Millipore Sigma | 250401 | 30 µM |
| Chemical compound, drug | [3H]GDP | PerkinElmer Life Sciences | NET966 | 3000 cpm/pmol |
| Chemical compound, drug | GTPgS35 | PerkinElmer Life Sciences | NEG030H | |
| Software, algorithm | ImageJ software | ImageJ (http://imagej.nih.gov/ij/) | ImageJ, RRID:SCR_003070 | |
| Software, algorithm | GraphPad Prism software | GraphPad Prism (https://graphpad.com) | GraphPad Prism, RRID:SCR_002798 | Version 8.0.0 |

## Protein purification and ARL3 GEF assay

Plasmids directing the expression of mouse GST-ARL13B or GST-ARL13B$^{V358A}$ proteins were transiently transfected into HEK cells and the recombinant proteins were later purified by affinity chromatography using glutathione-Sepharose, as described previously (*Cavenagh et al., 1994*; *Ivanova et al., 2017*). Human ARL3 (98.4% identical to mouse ARL3) was expressed in BL21 bacteria and purified as previously described (*Van Valkenburgh et al., 2001*). The ARL3 GEF assay was performed as previously described (*Ivanova et al., 2017*). Briefly, ARL3 (10 µM) was pre-incubated with [$^3$H]GDP (1 µM; PerkinElmer Life Sciences, specific activity ~3000 cpm/pmol) for 1 hour at 30℃ in 25 mM HEPES, pH 7.4, 100 mM NaCl, 10 mM MgCl$_2$. In contrast, pre-loading of ARL13B (10 µM) was achieved by incubation in 100 µM GTPγS for 1 minute at room temperature, due to its much more rapid exchange kinetics. The GEF assay was initiated upon addition of ARL3 (final = 1 µM), ARL13B (final = 0 or 1 µM), 10 µM GTPγS, and 100 µM GDP (to prevent re-binding of any released [$^3$H]GDP), in a final volume of 100 µL. The intrinsic rate of GDP dissociation from ARL3 was determined in parallel as that released in the absence of any added ARL13B. The reactions were stopped at different times (0–15 minutes) by dilution of 10 µL of the reaction mixture into 2 ml of ice-cold buffer (20 mM Tris, pH 7.5, 100 mM NaCl, 10 mM MgCl$_2$, 1 mM dithiothreitol). The amount of ARL3-bound [$^3$H]GDP was determined by rapid filtration through BA85 nitrocellulose filters (0.45 µm, 25 mm, Whatman), with 3 × 2 mL washes, and quantified using liquid scintillation counting. Time points are routinely collected in at least duplicate and each experiment reported was repeated at least twice, yielding very similar results. Data were analyzed in GraphPad Prism 7 software.

## Mouse allele generation and identification

All mice were cared for in accordance with NIH guidelines and Emory's Institutional Animal Care and Use Committee (IACUC). Lines used were *Arl13b$^{V358A}$* (*Arl13b$^{em1Tc}$*) [MGI: 6256969], *Arl13b$^{hnn}$* [MGI:

3578151] and FVB/NJ (Jackson Laboratory). To generate the V358A point mutation in the mouse, a CRISPR gRNA (CCAGTCAATACAGACGAGTCTA) targeting exon 8 of the *Arl13b* locus along with a donor oligo (5'-CCTATATTCTTCTAGAAAACAGTAAGAAGAAAACCAAGAAACTACGAA TGAAAAGGAGTCATCGGGCAGAACCAGTGAATACAGACGAGTCTACTCCAAAGAGTCCCACGCC TCCCCAAC-3'; underlined bases are engineered) were designed to generate a T-to-C change creating the V358A point mutation and C-to-G change creating a TspRI restriction site that could be used for genotyping (Millipore Sigma). The gRNA (50 ng/μl), oligo donor (50 ng/μl) and CRISPR protein (100 ng/μL) were injected by the Emory Transgenic and Gene Targeting core into one-cell C57Bl/6J zygotes. Zygotes were cultured to 2 cell before being transferred to pseudopregnant females. Genomic tail DNA from resulting offspring was amplified using PCR primers (5'-GAAGCAGGCA TGGTGGTAAT-3' and 5'-TGAACCGCTAATGGGAAATC-3') located upstream and downstream of the donor oligo breakpoints. The products were sequenced (5'-GAAGCAGGCATGGTGGTAAT-3') and 2 animals were identified heterozygous for the desired change and with no additional editing. One line (#173) was bred to FVB/NJ for three generations prior to performing any phenotypic analysis. Males from at least two distinct meiotic recombination opportunities were used in each generation that to minimize potential confounds associated with off-target CRISPR/Cas9 editing.

Genotyping was performed on DNA extracted from ear punch or yolk sac via PCR using the following primers: Fwd: 5'- AAGAATGAAAAGGAGTCAGCG −3', Rev: 5'- TGAACCGCTAA TGGGAAACT −3'; a SNP was engineered in the forward primer that, in combination with the V358A mutation, created a Cac8I restriction site. Thus, the PCR product was digested with Cac8I enzyme, run out on a 4% agarose gel and the relevant bands were detected: undigested wild type (~192 bp) and digested mutant bands (~171 bp).

## Phenotypic analysis of embryos

Timed matings of heterozygous intercrosses were performed to generate embryos of the indicated stage, with somite-matched pairs examined at each stage when appropriate. Embryos were dissected in cold phosphate-buffered saline and processed for immunofluorescence staining as previously described (*Horner and Caspary, 2011*).

## Mouse embryonic fibroblasts

Mouse embryonic fibroblasts (MEFs) were isolated and immortalized as previously described *Mariani et al. (2016)*. The identity of all cell lines were confirmed by genotyping PCR and tested mycoplasma negative. MEFs were maintained in DMEM with 10% fetal bovine serum (FBS) and 1% penicillin/streptomycin at 37°C in 5% $CO_2$. For experimental use, $Arl13b^{+/+}$, $Arl13b^{V358A/+}$, and $Arl13b^{V358A/V358A}$ and $Arl13b^{hnn/hnn}$ MEFs were grown on coverslips at a density of $0.5 \times 10^6$ cells/ mL and treated for 24 hours with 0.5% FBS Shh conditioned media or 0.5% FBS media to induce ciliogenesis (*Larkins et al., 2011*).

## Antibodies

Primary antibodies used were: mouse anti-Shh (5E1), mouse anti-Nkx2.2 (74.5A5), mouse anti-HB9 (81.5C10), mouse anti-Nkx6.1 (F55A10) (1:5 Developmental Studies Hybridoma Bank); rabbit anti-Olig2 (1:300 Millipore AB9610); mouse anti-acetylated α-tubulin (1:2500 Millipore Sigma; T6793), mouse anti-ARL13B (1:1000 NeuroMab N295B/66), rabbit anti-ARL13B (1:1000, Protein Tech 17711– 1-AP); rabbit anti-ARL13B sera from three distinct rabbits (503, 504 and 505) (Caspary Lab *Caspary et al., 2007*), rabbit anti-Smoothened (1:1000, kindly provided by K. Anderson); rabbit anti-IFT88 (1:1000, kindly provided by B. Yoder); rabbit anti-ARL3 (1:1000, *Cavenagh et al., 1994*); rabbit anti-INPP5E (1:150, Protein Tech 17797–1-AP); guinea pig anti-Gli2 (1:200, kindly provided by J. Eggenschwiler); rabbit anti-Ptch1 (1:150, kindly provided by R. Rohatgi), goat anti-Sufu (1:100, SC10933, Santa Cruz); goat anti-Gli3 (1:200, R and D AF3690); Alexa Fluor 488 and Alexa Fluor 568 (1:300, ThermoFisher); and Hoechst nuclear stain (1:3000). In all depicted images the red channel is false colored as magenta for the benefit of readers.

## Image quantification

Fluorescent intensities were measured in ImageJ software. Cilia were first identified by positive acetylated α-tubulin and an outline was hand drawn around the length of the cilium. The channel was

switched to the protein of interest and a measurement of the average fluorescent intensity was acquired. The same outline was then used to acquire a background reading of the cell-body that most closely matched the background at the cilium. In all cases immunofluorescent averages of proteins of interest in cilia were normalized to cell-body intensities (*Figure 2—figure supplement 2*). For samples with antibodies targeting Gli2, Gli3, and Sufu the ciliary tip was isolated and measured. The cilia tip was identified by weak acetylated α-tubulin staining or the cilium base was identified by the presence of the acetylated α-tubulin positive fibrils in the cell body that cluster at the base of the cilium (*Larkins et al., 2011*). For samples with antibodies targeting Smo, Ptch1, ARL3, and INPP5E the entire cilium was measured. We plotted the ratio of fluorescence intensity of the protein of interest to the cell body background as violin plots. Within each plot the dashed lines represent the median and interquartile range. The number of cilia examined per genotype and per condition are listed below their respective plot. All data, except for Smo which varied both genotype and treatment, were analyzed by one-way ANOVA. In the event of a significant ANOVA, Tukey's multiple comparisons were employed to determine significance as all groups were compared. Smo fluorescent intensity data were analyzed by two-way ANOVA. As only specific groups were compared, the significance of those comparisons across and between groups were made using Sidak's post-hoc to not inflate the significance of those comparisons Data were analyzed in GraphPad Prism 7 software.

In a similar experiment, retrograde IFT was inhibited by the addition of 30 μM ciliobrevin-D (Millipore Sigma; 250401) or 0.1% DMSO control in low-serum media for 60 minutes after ciliation was induced by serum starvation for 24 hours prior with 0.5% FBS media. Cilia were identified by staining with antibodies directed against acetylated α-tubulin, IFT88, and ARL13B. Fluorescent intensities were measured in ImageJ software. Again, cilia were first identified by positive acetylated α-tubulin and an outline was hand drawn. The channel was switched to the protein of interest and a measurement of the average fluorescent intensity was acquired. The same outline was then used to acquire a background reading of the cell-body. To measure IFT88, the cilia tip was identified by weak acetylated α-tubulin staining or the cilium base was identified by the presence of acetylated α-tubulin fibrils in the cell body that cluster at the base of the cilium (*Larkins et al., 2011*). To make comparisons among cell lines of distinct genotype with and without ciliobrevin-D treatment, data were analyzed by two-way ANOVA. To analyze the specific differences across groups, multiple comparisons were made using Sidak's post-hoc.

### Western blots

Western blotting was performed as previously described *Mariani et al. (2016)*, with antibody against ARL13B (1:1000 Neuromab N295B/66). Lysates were prepared with RIPA buffer and Sigma-Fast protease inhibitors (*Nachtergaele et al., 2013*). Values presented as volume intensity measured by chemiluminescence detected on ChemiDoc Touch Imaging System were normalized to total protein as measured on a stain-free gel (*Thacker et al., 2016*; *Rivero-Gutiérrez et al., 2014*). Normalized values were analyzed by one-way ANOVA and Tukey's multiple comparisons. Data were analyzed in GraphPad Prism 7 software.

## Acknowledgements

This work was supported by funding from NIH grants R01NS090029 and R35GM122549 to TC and R35GM122568 to RAK. EDG was supported by NIH training grant T32NS096050 and a diversity supplement to R01NS090029. Further support came from the Emory Integrated Mouse Transgenic and Gene Targeting Core, which is subsidized by the Emory University School of Medicine and with support by the Georgia Clinical and Translational Science Alliance of the National Institutes of Health under Award Number UL1TR002378 as well as the Emory University Integrated Cellular Imaging Microscopy Core of the Emory Neuroscience NINDS Core Facilities grant, P30NS055077. We are grateful to Alyssa Long and Sarah Suciu for sequencing potential CRISPR founders, and to members of the Caspary lab for discussion and manuscript comments.

## Additional information

### Funding

| Funder | Grant reference number | Author |
|---|---|---|
| National Institute of Neurological Disorders and Stroke | R01NS090029 | Tamara Caspary |
| National Institute of General Medical Sciences | R35GM122549 | Tamara Caspary |
| National Institute of General Medical Sciences | R35GM122568 | Richard A Kahn |
| National Institute of Neurological Disorders and Stroke | T32NS096050 | Eduardo D Gigante |
| National Institute of Neurological Disorders and Stroke | F31 NS106755 | Eduardo D Gigante |

The funders had no role in study design, data collection and interpretation, or the decision to submit the work for publication.

### Author contributions

Eduardo D Gigante, Formal analysis, Validation, Investigation, Visualization, Methodology; Megan R Taylor, Formal analysis, Investigation; Anna A Ivanova, Formal analysis, Validation, Investigation, Visualization; Richard A Kahn, Supervision, Funding acquisition, Visualization, Methodology; Tamara Caspary, Conceptualization, Supervision, Funding acquisition, Project administration

### Author ORCIDs

Eduardo D Gigante  https://orcid.org/0000-0002-1486-5377
Tamara Caspary  https://orcid.org/0000-0002-6579-7589

### Ethics

Animal experimentation: All mice were cared for in accordance with NIH guidelines and Emory's Institutional Animal Care and Use Committee (IACUC) under protocols DAR-2003545-072919N and PROTO201700587.

### Decision letter and Author response

Decision letter https://doi.org/10.7554/eLife.50434.sa1
Author response https://doi.org/10.7554/eLife.50434.sa2

## Additional files

### Supplementary files

• Transparent reporting form

### Data availability

All data generated or analysed during this study are included in the manuscript and supporting files.

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
