## [Decision Letter]

Thank you for submitting your article "ARL13B regulates Sonic Hedgehog signaling from outside primary cilia" for consideration by *eLife*. Your article has been reviewed by three peer reviewers, one of whom is a member of our Board of Reviewing Editors, and the evaluation has been overseen by Anna Akhmanova as the Senior Editor. The reviewers have opted to remain anonymous.

The reviewers have discussed the reviews with one another and the Reviewing Editor has drafted this decision to help you prepare a revised submission.

Summary:

This work builds on Mariani et al., 2016 which demonstrated that restricting *Arl13b* from localizing to cilia does not interfere with Hh signaling, but does interfere with Shh-stimulated chemotaxis. The previous work demonstrated that Arl13b-GFP mutants R200C and V358A do not localize in cilia, but are able to support normal Hh responses and SMO localization to cilia. The R200C mutant, as described by Nozaki et al., 2016, could be detected inside cilia (Figure 3F). The major advance in this manuscript is the creation and phenotypic analysis of a mouse *Arl13b^V358A^* allele. V358A excludes, as much as is measurable, ARL13B protein from the cilia. This work is a bit unusual in that the advances are based on negative results (absence of phenotypes and absence of localization).

Essential revisions:

Surprisingly, mice homozygous for the *Arl13b^V358A^* allele do not display overt developmental defects, suggesting that *Arl13b* functions outside the cilium to promote Hh signaling. The authors conclude in their title: "*Arl13b* regulates Sonic Hedgehog signaling from outside primary cilia." The major limitation to their conclusions is that a small amount of ARL13B may be in cilia and that is sufficient for SHH function and, as the authors acknowledge elsewhere in the manuscript, "It is formally possible that an extremely small amount of *Arl13b^V358A^* remains in cilia." As I'm sure the authors also realize, demonstrating that a protein does not localize to or function within a subdomain of the cell, especially an enzymatic protein that may operate at substoichiometric levels, is difficult. Although the expression level of *Arl13b^V358A^* appears to be only slightly reduced compared to Wt *Arl13b*, it is concerning that no Arl13B signal could be detected anywhere in neural tube of V358A/V358A embryos (Figure 2C). This raises the concern that the antibodies used do not sensitively detect *Arl13b^V358A^*. To make claims that *Arl13b^V358A^* does not function in the cilium, the authors should use an orthogonal approach, such as tethering wild type *Arl13b* outside the cilium in the null MEFs, to assess whether it can operate outside the cilium. Additionally, it would be good to provide some estimate of the lowest level of detectable protein in the cilium to provide a statement indicating that only if "x" percentage of ARL13B were ciliary and sufficient for SHH signaling, then the conclusions would not be valid.

Figure 1 is not sufficiently interpretable in its present form. In the present data, the kinetics of ARL3 GEF activity by wild type or V358A reaches maximum activity ~40 seconds into the time course. By adding excessive protein, the reaction is too rapid to show any quantitative differences in the activity of the wild type and mutant proteins. This experiment, as currently presented, shows that the mutant protein does have some activity, but could mask a 10-fold lower activity than wild type. The argument that the mutation is outside the homology defining the catalytic domain and thus unlikely to function only for targeting ignores the possibility of interdomain allostery/regulation, which is found in many small GTPases. For example, the C-terminus could fold over and inhibit the GTPase domain when engaged with its trafficking machinery, and this inhibition would be relieved to activate the GTPase when properly localized. The authors should compare various concentrations of wild type and mutant protein to see if the kinetic profiles are separable at lower concentrations.

The other source for the claim that *Arl13b^V358A^* cannot localize to cilia is the finding that ciliobrevin-D, a dynein inhibitor, does not cause the ciliary accumulation of *Arl13b^V358A^*. It appears that ciliobrevin D treatment may not be efficiently blocking retrograde IFT as the increase of IFT88 in the majority of the ciliobrevin D-treated cells is subtle. Moreover, it is not clear that all ciliary proteins leave cilia in a dynein-dependent way. Nor is it clear that dynein is functional inside cilia of cells expressing *Arl13b^V358A^* as Arl3 localization is decreased and Arl3 is important for dynactin and cargo unloading from dynein (Jin et al., Nature Communications 2014). Some of these limitations should be included in the Discussion.

The authors demonstrate that Arl3 localization is compromised in *Arl13b^V358A^*^-^expressing mice. As Arl3 is thought to function at cilia, the prediction is that *Arl13b^V358A^* mice would possess phenotypes similar to Arl3 loss of function. The authors do a nice job of assessing Hh-associated phenotypes. Do the mutant homozygous mice possess Arl3-associated phenotypes such as renal cysts and retinal degeneration? If not, do the authors propose that Arl3 also functions outside of the cilium?

---

## [Author Response]

Essential revisions:Surprisingly, mice homozygous for the Arl13bV358A allele do not display overt developmental defects, suggesting that Arl13b functions outside the cilium to promote Hh signaling. The authors conclude in their title: "Arl13b regulates Sonic Hedgehog signaling from outside primary cilia." The major limitation to their conclusions is that a small amount of ARL13B may be in cilia and that is sufficient for SHH function and, as the authors acknowledge elsewhere in the manuscript, "It is formally possible that an extremely small amount of Arl13bV358A remains in cilia." As I'm sure the authors also realize, demonstrating that a protein does not localize to or function within a subdomain of the cell, especially an enzymatic protein that may operate at substoichiometric levels, is difficult. Although the expression level of Arl13bV358A appears to be only slightly reduced compared to Wt Arl13b, it is concerning that no Arl13B signal could be detected anywhere in neural tube of V358A/V358A embryos (Figure 2C). This raises the concern that the antibodies used do not sensitively detect Arl13bV358A.

We appreciate the reviewers’ recognition that we acknowledge subdetectable amounts of ARL13B may be present in *Arl13b^V358A/V358A^*cilia. As to the concern regarding ARL13B signal in the neural tube, we guide the reviewers to the overexposed images of neural tubes stained for ARL13B and IFT88 in Figure 2—figure supplement 1. Upon overexposure, where the cilia staining of wildtype cells is saturated, we do observe cell body staining in *Arl13b^V358A/V358A^*neural tube that is above background levels detected in *Arl13b^hnn^* neural tube. We note that we do not detect any ARL13B in cilia in these images. In the images to which the reviewer refers in Figure 2, we calibrated to avoid saturation of the cilia in *Arl13b^+/+^* and *Arl13b^V358A/+^*samples. Therefore, we acquired the *Arl13b^V358A/V358A^*neural tube images under the same exposure parameters and conditions.

These data suggest that the cellular pool of ARL13B is difficult to detect. To explore this further, we examined ARL13B and acetylated a tubulin staining in *IFT172^wim^* MEFs which lack cilia. As controls we used wildtype, *Arl13b^hnn^* (lacking ARL13B), and *IFT172^wim^ Arl13b^hnn^* (lacking ARL13B and cilia) MEFs. As expected, we found acetylated a tubulin stained cilia only in wildtype and *Arl13b^hnn^* MEFs, and no ARL13B positive cilia in *Arl13b^hnn^* cells. Like in the neural tube, we took pictures using exposure levels that avoided saturation of the ARL13B channel (which we call normal exposure) and at 4X the normal exposure level, where the ARL13B channel is saturated in the cilium (which we call 4X or over exposed). At normal exposure, we do not detect signal above background staining in the cell body of wildtype, *IFT172^wim^*, *Arl13b^hnn^*, and *IFT172^wim^ Arl13b^hnn^* cells. At 4X exposure, we observe ARL13B throughout the cytoplasm of wildtype and *IFT172^wim^* cells, the signal is modestly higher than the background level that we detect in *Arl13b^hnn^* and *IFT172^wim^ Arl13b^hnn^* controls. Taken together, these data indicate that cellular ARL13B is just at the cusp of being detectable- but is present. We modified the text to reflect these points and added the MEF data as Figure 2—figure supplement 2.

To make claims that Arl13bV358A does not function in the cilium, the authors should use an orthogonal approach, such as tethering wild type Arl13b outside the cilium in the null MEFs, to assess whether it can operate outside the cilium.

We note that this result is consistent with previous work using a D1R-RAB23^S23N^ construct which greatly diminished, but did not ablate, DR’s ciliary localization (Figure 7, Leaf et al., 2015). Additionally, ARL13B is used to target proteins to cilia so this result indicates ARL13B is capable to target RAB23^S23N^ to the cilium.

As the reviewers suggested, tethering ARL13B outside the cilium would be a clever approach to bolster our claim. To this end, we generated a wild type ARL13B-RAB23^S23N^ fusion protein construct. The GTP-binding mutant RAB23^S23N^ disrupts RAB23 cilia localization, as well as proteins to which it is fused. We expressed either ARL13B-RAB23^S23N^ or ARL13B-GFP in *Arl13b^hnn^* cells and found, like ARL13B-GFP, ARL13BRab23^S23N^, localized in cilia (see Author response image 1. This indicates that we are not technically able to tether wild type ARL13B outside the cilium at this time.

**Author response image 1. respfig1:** ARL13B-RAB23^S23N^ fusion protein is present in the cilia of *Arl13b^hnn^* MEFs. *Arl13b^hnn^* MEFs transfected with ARL13B-RAB23^S23N^, ARL13B-GFP, or control plasmid. (**A**) The ARL13BRAB23 ^S23N^ fusion protein is detectable in the cilium (acetylated a-tubulin, magenta) by ARL13B antibody (cyan, NeuroMab). (**B**) Wildtype ARL13B-GFP is present in the cilium in high- (left column) and low- (right column) expressing cells. (**C**) No ARL13B positive cilia were detected in control plasmid transfected cells. (**D**) Secondary only controls.

Additionally, it would be good to provide some estimate of the lowest level of detectable protein in the cilium to provide a statement indicating that only if "x" percentage of ARL13B were ciliary and sufficient for SHH signaling, then the conclusions would not be valid.

We understand the desire to ascribe a clear quantitative measurement to the amount of ciliary ARL13B that might exist in *Arl13b^V358A/V358A^* cilia. However, as the reviewers predicted, we are acutely aware of the challenges in doing so with, “an enzymatic protein that may operate at substoichiometric levels”. In reality, there is no way to know whether any estimate could be biologically meaningful. Rather, what our data clearly demonstrate that the cilia defects that we previously interpreted to be causing defective Shh signal transduction, persist in the *Arl13b^V358A/V358A^* mice. We have modified the text to highlight this point.

Figure 1 is not sufficiently interpretable in its present form. In the present data, the kinetics of ARL3 GEF activity by wild type or V358A reaches maximum activity ~40 seconds into the time course. By adding excessive protein, the reaction is too rapid to show any quantitative differences in the activity of the wild type and mutant proteins. This experiment, as currently presented, shows that the mutant protein does have some activity, but could mask a 10-fold lower activity than wild type. The argument that the mutation is outside the homology defining the catalytic domain and thus unlikely to function only for targeting ignores the possibility of interdomain allostery/regulation, which is found in many small GTPases. For example, the C-terminus could fold over and inhibit the GTPase domain when engaged with its trafficking machinery, and this inhibition would be relieved to activate the GTPase when properly localized. The authors should compare various concentrations of wild type and mutant protein to see if the kinetic profiles are separable at lower concentrations.

We completely agree with this reviewer that by measuring ARL3 GEF activity using a single concentration of ARL13B (or mutant) one cannot conclude that no change to affinity for substrate results from mutation. And we did not make such a claim; rather, only that the mutant retains activity. We have gone over the entire manuscript with a renewed resolve to ensure that a reader not come away with the mistaken impression that we make such a claim. We have previously used the same approach described in this submission to test a series of 7 other designed mutations in ARL13B (Ivanova et al., 2017) and a human patient mutation (Rafiullah et al., 2017). Note that in our assays, we are testing ARL3 GEF activities using equimolar substrate (ARL3) and “enzyme” (ARL13B). This is consistent with a single turnover catalytic event and poor sensitivity in the assay. In addition, because there is spontaneous release of GDP during the course of the assay incubation, the GEF (ARL13B)-stimulated release has a narrower window than for standard enzyme assays. We also want to point out that in the initial demonstration of ARL3 GEF activity by ARL13B (Gotthardt et al., 2015) in which a number of ARL13B mutants were examined, they too used a single concentration of ARL13B but in this case it was 10-fold (5μM) over that of the substrate (ARL3, 0.5 μM). In only one figure in that paper did they report the effects of different concentrations of ARL13B on the GEF activity and they show that an increase of 10-fold (0.5 vs. 5.0 μM) resulted in less than a 50% change in rate of GDP release. We are aware of the limitations imposed on interpretations resulting from the lack of detailed dose response curves. Unfortunately, for a number of technical reasons we have found that attempts to generate such dose response curves are problematic and not publishable. We hope this clarifies the situation and reviewers better understand the limitations imposed upon us by this assay, which although not robust in turnover, we believe is a valid way to assess whether a point mutation results in substantial changes to activity.

The other source for the claim that Arl13bV358A cannot localize to cilia is the finding that ciliobrevin D, a dynein inhibitor, does not cause the ciliary accumulation of Arl13bV358A. It appears that ciliobrevin D treatment may not be efficiently blocking retrograde IFT as the increase of IFT88 in the majority of the ciliobrevin D-treated cells is subtle.

The effect is subtle but present. We found that treating cells with ciliobrevin-D over a longer time period or with higher concentrations resulted in ciliary disassembly, consistent with it efficiently blocking retrograde IFT. Thus, the 60 minute timepoint was as long as we could leave the cells and still observe cilia. To corroborate the IFT88 result, we repeated the experiment using GLI3. GLI3 is normally transported via IFT and accumulates at the ciliary tip upon retrograde transport being blocked. We found a defined increase in GLI3 at the ciliary tip following treatment with ciliobrevin-D. The magnitude is on par with our previous finding with IFT88. We did not detect ciliary *ARL13B^V358A^*. We include these data and analysis in Figure 3, and amended the manuscript.

Moreover, it is not clear that all ciliary proteins leave cilia in a dynein-dependent way. Nor is it clear that dynein is functional inside cilia of cells expressing Arl13bV358A as Arl3 localization is decreased and Arl3 is important for dynactin and cargo unloading from dynein (Jin et al., Nature Communications 2014). Some of these limitations should be included in the Discussion.

The reviewers are correct that ARL13B movement in cilia may not be regulated by dynein motors and therefore would be unaffected by ciliobrevin-D treatment. The suggestion (below) to disrupt IFT27 function to address this possibility would be appropriate if we had the applicable reagents. As requested, we now discuss this possibility in the Discussion.

The authors demonstrate that Arl3 localization is compromised in Arl13bV358A-expressing mice. As Arl3 is thought to function at cilia, the prediction is that Arl13bV358A mice would possess phenotypes similar to Arl3 loss of function. The authors do a nice job of assessing Hh-associated phenotypes. Do the mutant homozygous mice possess Arl3-associated phenotypes such as renal cysts and retinal degeneration? If not, do the authors propose that Arl3 also functions outside of the cilium?

As indicated in the Materials and methods, we maintain *Arl13b^V358A/V358A^* mice on an FVB/NJ background, which carry the *Pde6^brd1^* mutation that causes retinal degeneration and blindness by 3 weeks of age (Chang et al., 2002; Wong et al., 2006). We are crossing the mice to an appropriate background to examine the retinal phenotype. We do have some preliminary observations that *Arl13b^V358A/V358A^* mice display kidney cysts so we do not currently have any data to support ARL3 functioning from outside the cilium.

**References:**

Chang, B, NL Hawes, RE Hurd, MT Davisson, S Nusinowitz, and JR Heckenlively. 2002. Retinal degeneration mutants in the mouse, Vision research, 42: 517-25.

Gotthardt, K., M. Lokaj, C. Koerner, N. Falk, A. Giessl, and A. Wittinghofer. 2015. A G-protein activation cascade from Arl13B to Arl3 and implications for ciliary targeting of lipidated proteins, eLife, 4.

Ivanova, A. A., T. Caspary, N. T. Seyfried, D. M. Duong, A. B. West, Z. Liu, and R. A. Kahn. 2017. Biochemical characterization of purified mammalian ARL13B protein indicates that it is an atypical GTPase and ARL3 guanine nucleotide exchange factor (GEF), J Biol Chem, 292: 11091-108.

Leaf, Alison, and Mark Von Zastrow. 2015. Dopamine receptors reveal an essential role of IFTB, KIF17, and Rab23 in delivering specific receptors to primary cilia, eLife, 4: e06996.

Rafiullah, R., A. B. Long, A. A. Ivanova, H. Ali, S. Berkel, G. Mustafa, N. Paramasivam, M. Schlesner, S. Wiemann, R. C. Wade, E. Bolthauser, M. Blum, R. A. Kahn, T. Caspary, and G. A. Rappold. 2017. A novel homozygous ARL13B variant in patients with Joubert syndrome impairs its guanine nucleotide-exchange factor activity, Eur J Hum Genet, 25: 1324-34.

Wong, AA, and RE Brown. 2006. Visual detection, pattern discrimination and visual acuity in 14 strains of mice, Genes, Brain and Behavior, 5: 389-403.